# Designed Rubredoxin miniature in a fully artificial electron chain triggered by visible light

Marco Chino [1], Luigi Franklin Di Costanzo[2], Linda Leone [1], Salvatore La Gatta [1], Antonino Famulari [3,4], Mario Chiesa [3], Angela Lombardi [1] ✉ & Vincenzo Pavone [1] ✉

Designing metal sites into de novo proteins has significantly improved, recently. However, identifying the minimal coordination spheres, able to encompass the necessary information for metal binding and activity, still represents a great challenge, today. Here, we test our understanding with a benchmark, nevertheless difficult, case. We assemble into a miniature 28-residue protein, the quintessential elements required to fold properly around a $FeCys_4$ redox center, and to function efficiently in electron-transfer. This study addresses a challenge in de novo protein design, as it reports the crystal structure of a designed tetra-thiolate metal-binding protein in sub-Å agreement with the intended design. This allows us to well correlate structure to spectroscopic and electrochemical properties. Given its high reduction potential compared to natural and designed $FeCys_4$-containing proteins, we exploit it as terminal electron acceptor of a fully artificial chain triggered by visible light.

Electron transport chains play a central role in many life-sustaining functions from respiration[1,2], to light harvesting[3,4]. They involve two or more redox-active metalloproteins, with one or more metal cofactors bound in their interior. These metal cofactors are highly conserved in their first coordination sphere, and the surrounding residues intimately modulate their electronic structure. A wide range of reduction potentials can be achieved, thus generating the driving force of electron cascades. The protein matrix also drives the mutual orientation of these cofactors, by subtly evolved self-assembly processes, fundamentally regulating electron-transfer. Thus, it is imperative in metalloprotein design to develop finely tunable redox-active metal sites, amenable for photo-induced electron trafficking and bioenergy control. Previous work has been focused on charge-separation/recombination at purposely optimized abiotic cofactors[5,6], electron transfer towards natural acceptors[7,8], injection into titanium-based photoanodes[9], as well as intra-protein electron transfer between two different cofactors[10,11]. Among others, the synthetic metalloporphyrin-containing proteins,

named Mimochromes (MC)[12], previously developed by us, have been already exploited in electron transfer and may be tuned for their use as photosensitizers. Indeed, the best-performing model, MC6*a, is able to host several metal ions (Fe, Mn, Co), displaying different activities[12–14].

In nature, most of the redox proteins involved in electron trafficking and bioenergy control are represented by cupredoxins[15,16], cytochromes[17,18], and iron-sulfur proteins[19–21]. Rubredoxins (Rds) represent the simplest and most studied case (Fig. 1). They bind a single iron ion through four Cys Sγ with an almost tetrahedral geometry and they can cycle between the oxidation states (II) and (III). Rds (45–55 amino acids) adopt a $C_2$-pseudo-symmetric fold constituted by two symmetry-related CXXCX α-turns[22]. Despite well-conserved backbones and sequences (50–60% sequence identity), their reduction potential varies in the range −100/+50 mV in prokaryotes and could reach 125 mV (vs SHE, Standard Hydrogen Electrode) in eukaryotes[19]. Even higher reduction potential has been found for the rubredoxin-like domain of *Desulfovibrio vulgaris* rubrerythrin, a

[1]Department of Chemical Sciences, University of Naples Federico II, Via Cintia 21, 80126 Napoli, Italy. [2]Department of Agricultural Sciences, University of Naples Federico II, Via Università 100, 80055 Portici, Italy. [3]Department of Chemistry, University of Torino, Via Giuria 9, 10125 Torino, Italy. [4]Department of Condensed Matter Physics, University of Zaragoza, Calle Pedro Cerbuna 12, 50009 Zaragoza, Spain. ✉e-mail: alombard@unina.it; vipavone@unina.it

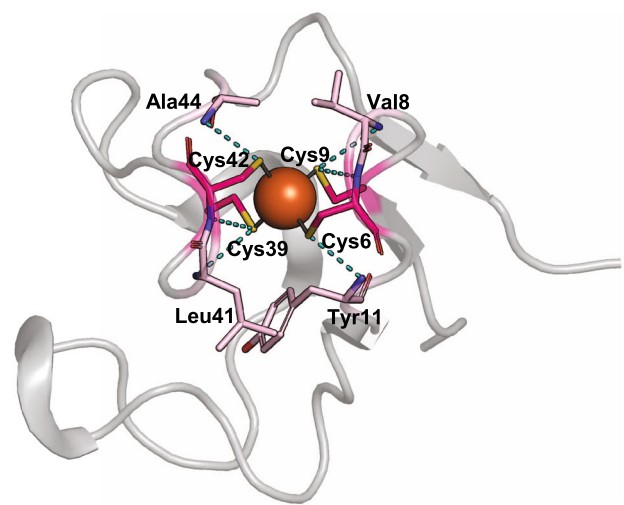

**Fig. 1 | Crystal structure of V44A mutant of *Cp* Rd (PDB ID: 1C09)[24].** The secondary structure is depicted as a gray ribbon, the iron ion as a brown sphere and the first (magenta) and second (pink) coordination sphere residues as sticks.

non-heme di-iron proteins belonging to the ferritin-like superfamily[19]. Mutagenesis studies have dissected the role of the second coordination sphere in modulating Rds potential[20,23–25], and some double mutants have shown that the effect of mutations is generally additive[26].

In this respect, several studies of rational redesign and fully de novo design have targeted the Rd system. Among these, some groups have focused on alternative metal ions, using the S₄ site as a surrogate of more complex catalysts, such as [NiFe] hydrogenases or molybdoenzymes[25,27]. Others have installed the tetrahedral FeCys₄ site in structurally different natural and de novo proteins[28–30]. We, and others, have focused on the Rd prototypical structural unit, making use of its intrinsic symmetry to build a miniaturized peptide scaffold[31–34]. In particular, we previously recognized the designed protein METP (Miniaturized Electron Transfer Protein) as a minimal unit needed to reproduce Rds by retrostructural analysis[33]. METP consists of two short undecapeptides, self-assembled around a tetrahedrally coordinating metal ion, and related by a twofold axis, as in Rd. Despite METP spectroscopic characterization indicated the expected structural arrangement when coordinated to different metal ions, its iron complex was unable to perform reversible redox cycles. An auto-redox reaction may account for the observed instability of the Fe(III)-tetrathiolate complex, with Fe(III) reduction to Fe(II), and disulfide formation.

In this work, we describe the design and characterization of a single-chain high-potential miniaturized electron transfer protein (named METPsc1), encompassing the FeCys₄ metal cofactor. We overcome three difficult challenges in de novo metalloprotein design. First, we implant a FeS₄ site into a de novo protein, made up of half the residues compared to natural Rds (28 *vs* ~55 residues), and closely matching the highest reported reduction potentials in the Rd family; secondly, we obtain the first X-ray structure of a tetra-thiolate metalloprotein designed from scratch, within sub-Å agreement with the intended design; thirdly, and most important, we establish a fully artificial electron chain triggered by visible light, exploiting the newly developed protein as terminal electron acceptor. The photosensitizer unit (ZnMC6*a) used in this process is itself an artificial protein, belonging to MCs. Taken together, our results demonstrate that such miniaturized proteins might be exploited in optoelectronics and light-harvesting biodevices, and open new perspectives to study more complex electron transfer chains. The METPsc1 small scaffold may offer a great opportunity for easily engineering the second coordination sphere amino acids, thus finely modulating the redox potential of different metal ions in sulphur-rich environments.

## Results

### Design strategy of a single-chain miniaturized FeS₄ protein

In recent studies, the introduction of asymmetry has been recognized as a key strategy for achieving or improving functions in designed metalloproteins[31,32,35–37]. With this respect, we generated asymmetry in our previous dimeric METP scaffold, by designing a single-chain peptide. Following the early METP design, we generated new backbone coordinates by miniaturization and symmetry considerations. We used as template the high-resolution structure of the reduced V44A mutant of *Cp* (*Clostridium pasteurianum*) Rd, which represents one of the high-potential mutants[24].

Figure 1 illustrates its secondary structure and highlights the first and second coordination sphere residues (PDB ID: [1C09])[24]. Starting from this structure (Fig. 2a), the segment from Val38 to Glu50 was dissected from the protein, and the $C_2$ longitudinal axis was applied (Fig. 2b) to generate the dimer coordinates. We then performed a systematic search to find the best fragment linking N- (Val38) and C-termini (Glu50 of the symmetric copy), fixing seven residues as the maximum gap length. We plotted the number of fragments within 1 Å backbone RMSD against the gap length (Supplementary Fig. 1), and we found that a 4-residues loop represented the shortest yet designable choice to link the two ends (39 hits out of 158 total hits, Fig. 2c).

As expected for a 4-residues segment, simple β-turn motifs were found in most cases (29 out of 39, Supplementary Table 1). The sequence analysis of the matches revealed that both $i+1$ and $i+2$ positions were frequently occupied by Gly residues (Supplementary Fig. 2), as typically observed in type I'/III' β-turns[38]. Type I' and III' β-turns were found in 14 and 3 structural hits, respectively (Supplementary Table 1). Interestingly, this search allowed us to overcome some of the limitation previously encountered to covalently link the two symmetry-related moieties, such as the use of stabilizing long β-hairpins[31] or synthetically difficult cyclization steps[32]. The best matching fragment was used to generate an initial backbone model, by grafting the loop coordinates onto the previously generated Val38-Glu50 $C_2$-symmetric dimer (Fig. 2d). This structure was then submitted to a preliminary flexible backbone design routine (see Supplementary Methods). This step helped identify some key features in terms of residue propensities at specific positions (Supplementary Fig. 3). Moreover, in this stage we fixed 2-aminoisobutiric (Aib) residues at *pseudo*-symmetric positions 9 and 24 to induce the 3₁₀-helix formation[39], as previously accomplished in METP design[33].

In a second design round (Supplementary Fig. 4), we instructed the side chain packing software routine with the results from the previous steps, and we further defined the identities of the X residues of the CXXC motif (Fig. 2e), by limiting them only to hydrophilic residues. The final designed model was obtained by further Monte Carlo sampling of the conformational subspace (300 runs) and selecting the structure with the lowest energy score (Fig. 2e, and Supplementary Methods).

The designed model is a compelling collection of secondary and super-secondary motifs, all of them collapsed into one small polypeptide chain (Fig. 2 and Supplementary Fig. 6). The *pseudo-twofold* symmetry axis is relating two consecutive similar segments formed by a progression of: (1) a small extended 2-residue β-strand; (2) an α-turn with Ser-Asp-Cys as corner residues; (3) Gly β-bulge; (4) a small extended 2-residue β-strand; (5) an incipient 3₁₀-helix with two consecutive β-turns; (6) a small extended 2-residue β-strand; (7) a type I' β-turn involving two consecutive Gly residues. Interestingly, β-strands pair to give two sets of short antiparallel β-sheets.

### ZnMETPsc1 crystal structure analysis

The newly designed METPsc1 miniprotein was synthesized in good yield by standard solid-phase methods (see Supplementary Information for details) and characterized by X-ray diffraction analysis as zinc complex at high resolution. ZnMETPsc1 crystallizes in the

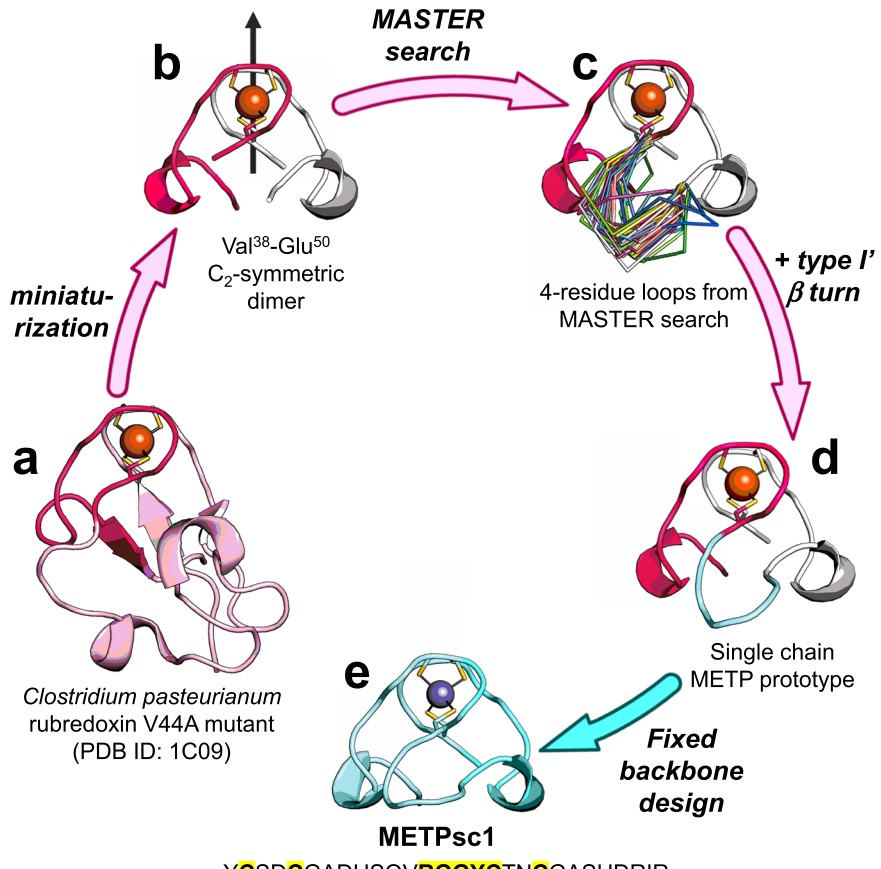

**Fig. 2 | Design of METPsc1. a** crystal structure of *Cp* Rd V44A mutant (PDB ID: 1C09). **b** miniaturized model, obtained by applying a $C_2$ longitudinal rotation to the Val38-Glu50 fragment of Cp Rd V44A. **c** superimposition of the 4-residue loops found from the fragment search. **d** single-chain METP prototype, obtained by combination of the $C_2$-symmetric dimer with the type I' β-turn selected from the search. **e** designed model and sequence of METPsc1, in its complex with $Zn^{2+}$. Cys and type I' β-turn residues are highlighted in yellow.

orthorhombic space group C222$_1$. The asymmetric unit of the cell contains one monomer. All protein residues were clearly identified from the electron density map and correspond to the designed protein sequence, including the N- and C-terminal acetyl and amide protecting groups, respectively (Fig. 3a, Supplementary Table 2). Notably, the comparison of the designed METPsc1 and its experimental X-ray structure (Fig. 3b, backbone RMSD 0.45 Å) are significantly similar, including the coordinative Cys residues and the hydrophobic sidechain packing. Surface-exposed sidechains adopt alternative rotamers, probably due to crystal packing and solvation interactions. When the minimized X-Ray structure was compared to the relaxed design models, RMSDs cluster at values ≥0.4 Å, which may be ascribed to some limitations in the metal binding scoring/constrains of the Rosetta energy function (Supplementary Results and Supplementary Fig. 5).

Noteworthy, all the designed secondary and super-secondary structural motifs are found in the experimental structure (Fig. 3c, Supplementary Information, Supplementary Fig. 6 and Supplementary Table 3). The peptide chain folds as a truncated cone shaped molecule (Supplementary Fig. 7), with an upper base corresponding to the metal binding site near the surface formed by Cys20-Asp4 and Cys5-Asn19 residues. The hydrophobic residues Aib9, Val12, Aib24 and Ile27, facing each other, with sidechains nearly aligned on a plane, form the lower base. Notably, Cys2 and Cys17, the other two cysteine residues completing the coordination sphere, occupy the innermost space of the whole protein. All the remaining residues decorate the external surface of the conical shape, forming a highly hydrophilic surface.

The shell around the macromolecules is hydrated and the crystal packing is characterized by interactions involving symmetrically related Arg residues. The crystal packing is stabilized by intermolecular salt bridges between a crystallographic related residue of Arg26 and Asp4 (see below). In addition, interactions between Tyr16 and the equivalent residue of a crystallographic related METPsc1 molecule are observed with a 3.32 Å distance between -OH atom groups.

## First and second sphere interactions define zinc complex features

$Zn^{2+}$ is tetrahedrally coordinated by four Cys Sγ with average Sγ-Zn distance of 2.34 ± 0.03 Å and Sγ-Zn-Sγ bond angle of 109 ± 4° (Fig. 3d), consistently with the geometry found in the twelve ultrahigh resolution rubredoxin structures retrieved from the Protein Data Bank (PDB)[40] that contain $Zn^{2+}$. The Cys residues are arranged around the metal center with a clockwise distribution of sidechains in that $\chi^1$ are either $g+$ or $t$ for Cys5/Cys20 and Cys2/Cys17, respectively. The torsion angle Sγ(Cys2)-Zn-Sγ-Cβ(Cys17) is 180° while the *pseudo*-symmetry-related torsion angle is Sγ(Cys6)-Zn-Sγ-Cβ(Cys20) is 159°.

The second coordination shell is characterized by H-bonds involving Cys Sγ and backbone N-H donors, similarly to natural Rds (Supplementary Table 4). Cys2 accepts H-bonds from backbone amide groups of Asp4 and Cys5, the same occurring for the symmetry related Cys17 (Asn19 and Cys20 backbone amides).

The designed sequence presents Ala residues at positions 7 and 22 (see sequence in Fig. 2e), being sufficiently small to let their own backbone N-H to H-bond Cys5 and Cys20 Sγ, respectively (Fig. 3e). The strength of this H-bond has previously been correlated to the

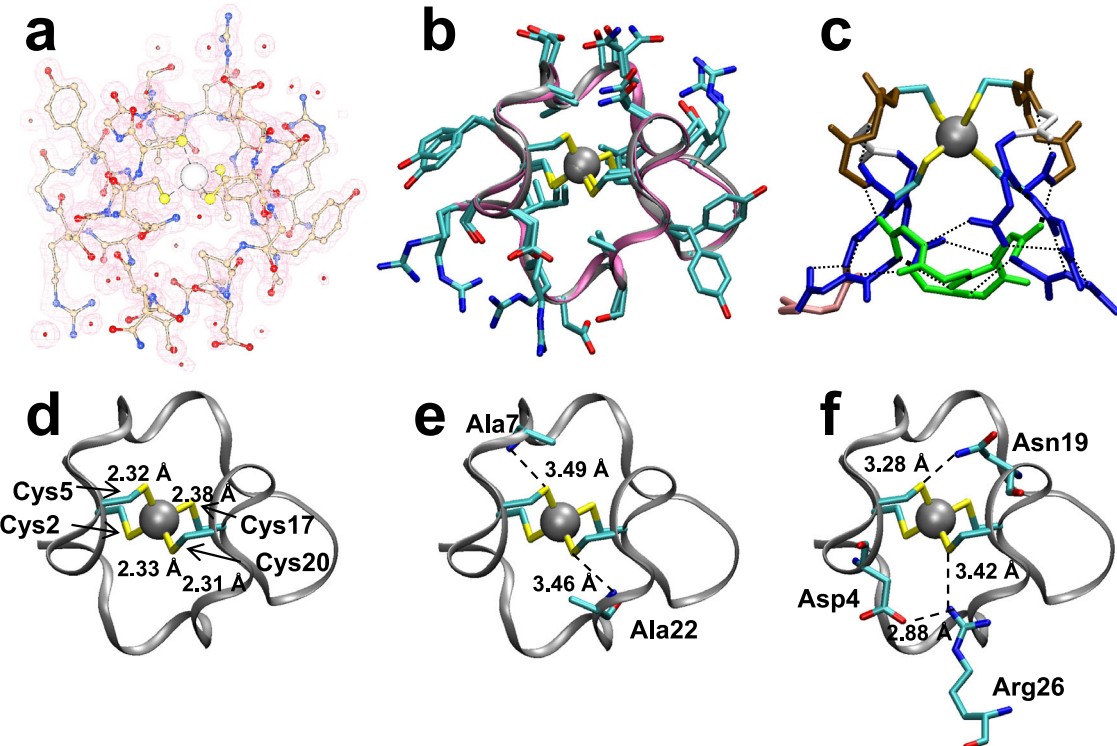

**Fig. 3 | ZnMETPsc1 structural characterization. a** Metal ion, all sidechains, and N- and C-terminal capping groups are clearly visible in the electron density map ($2F_o$-$F_c$. map, 1.3 $\sigma$ level). **b** The monomeric X-ray structure of ZnMETPsc1 (cyan, this work, PDB ID: 5SBG) closely matches the designed model (light brown). **c** Description of secondary structural elements found in ZnMETPsc1 structure (blue: β-strand; ocre: α-turn; gray: β-bulge; green: 3₁₀-helix; orange: type I' β-turn). Dashed lines represent backbone to backbone H-bonds. **d** First coordination sphere shows the expected coordination bond distances between zinc and cysteine sulfur atoms. **e** Second coordination sphere involving amide of Ala7 and Ala22 exacerbates H-bond strength with respect to wt *Cp* Rd. **f**, The H-bond donors from sidechains of Asn19 and a symmetry-related Arg26 (in cyan) to METPsc1 partners are indicated.

**Table 1 | Spectroscopic parameters of FeMETPsc1 and *Cp* Rd in Fe(II) and Fe(III) oxidation states**

|  |  | Fe²⁺ METPsc1 | Fe²⁺ *Cp* Rd | Fe³⁺ METPsc1 | Fe³⁺ *Cp* Rd |
|---|---|---|---|---|---|
| UV-Vis | λ/nm (ε/mM⁻¹ cm⁻¹) | 311 (7.73), 331 (4.43) | 311 (10.8), 333 (6.3)[42] | 345 (7.28), 370 (8.33), 494 (6.54), 570 (3.13), 745 (0.33) | 350 (7.00), 380 (7.70), 490 (6.60), 570 (3.20), 750 (0.35)[43] |
| CD | λ/nm (+/−) | 312(−), 333(+) | 314(−), 335(+)[44] | 437(+), 502(−), 557(+), 632(−) | 437(+), 500(−), 560(+), 635(−)[44] |
| EPR | $g_{eff}$ | a | a | 9.15, 4.26 | 9.4, 4.3[42] |

ªAlthough the high-spin (S = 2) ferrous iron is a paramagnetic species its integer spin state makes it usually difficult to detect under standard experimental conditions. Spin-Hamiltonian parameters have been measured by means of high-frequency EPR (HFEPR, ν ≥ 95 GHz) in references[68,69]. An effective $g_z$ = 2.08 ± 0.01 has been reported from X- and Q-band EPR studies for a variant of *Cp* Rd[70]. Due to low intensity and the large line widths involved, this signal was not observed in our experiments.

reduction potential, as shown for *Cp* Rd mutants of Val44[23]. Moreover, positions 4 and 19 of METPsc1 (Fig. 3f) correspond to position 41 of *Cp* Rd, the latter being crucial for the solvent accessibility and H-bonding of Cys9 in *Cp* Rd[41]. In our model, it is reasonable to hypothesize that Asp4 residue would drive water access towards Cys20. Asn19 residue donates its sidechain amide protons to Cys5 Sγ, further decreasing its electron density. Cys20 Sγ accepts a H-bond from sidechain guanidine group of a crystallographically related Arg26, mimicking a water molecule as observed in L41A *Cp* Rd X-ray structure (Fig. 3f).

**Structure/function correlations in FeMETPsc1**

Spectroscopic and electrochemical studies were performed to analyze the METPsc1 behavior in solution and to correlate structural to functional properties. Iron binding and coordination geometry were assessed by a combination of UV-Vis absorption, CD, and EPR spectroscopies (Table 1)[42–44].

METPsc1 forms a 1:1 complex with Fe²⁺ at pH 6.8, as assessed by Mohr salt titration of the apo peptide, under inert atmosphere (Fig. 4a). The data were well described by a binding isotherm with an apparent $K_D$ ≤ 300 nM. Such value is dramatically lower than those we previously observed for the dimeric METP (one and two order of magnitude, for Zn²⁺ and Co²⁺, respectively), most likely attributable to the enhanced chelate effect granted by the monomeric protein. METPsc1 is a tighter ligand for iron when compared to other previously designed monomeric constructs[30,31], but still looser than a previously reported zinc-finger inspired cyclic scaffold[32].

When exposed to air, Fe²⁺ complex readily oxidizes to the ferric state. We collected UV-Vis and CD spectra of both reduced and oxidized forms. Absorption spectra for both oxidation states show the Rd characteristic LMCT bands of tetrahedral thiolate donors (Fig. 4b). In addition, their extinction coefficients are in striking agreement with those reported for *Cp* Rd (Table 1). CD positive and negative Cotton effects alternate as previously reported for the ferric state[44] and lead to the assignment of at least six transitions in the visible region (Fig. 4c), four of which match those found in *Cp* Rd (Table 1), and in other designed models[30,32].

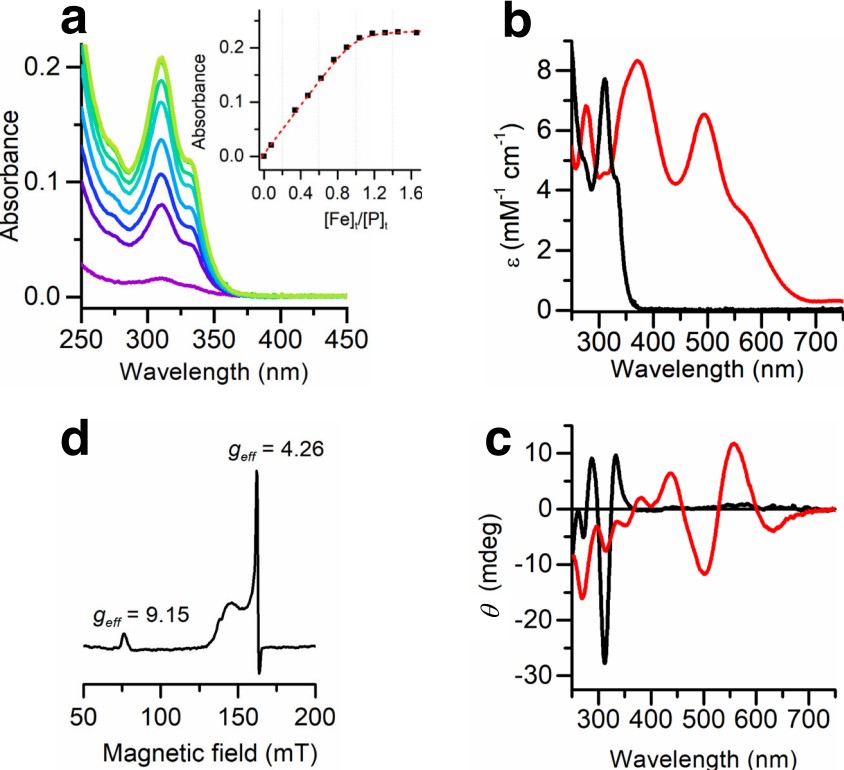

**Fig. 4 | FeMETPsc1 spectroscopic characterization. a** UV-Vis titration of METPsc1 with $Fe^{2+}$, spectra at increasing iron concentration are reported from violet to green. Absorbances at 311 nm are reported in the inset (black squares) and fitted by a 1:1 binding isotherm (red dashed line). Mohr's salt (36 mM) aliquots were added to a 30 μM METPsc1 solution in a 20 mM HEPES buffer (pH 7) and 1 mM TCEP. **b, c** UV-Vis and CD spectra of the reduced (black line) and oxidized (red line) FeMETPsc1 (40 μM) species. **d** X-band CW-EPR spectrum of $Fe^{3+}$METPsc1 (0.5 mM) in 20 mM phosphate buffer (pH 7) and 5 mM TCEP at 4.5 K. Source data are provided in a Zenodo repository under accession code 7748883.

The complex was also characterized by X-band Continuous Wave (CW)-EPR spectroscopy (Fig. 4d). The observed resonances, $g_{eff} = 9.15$ and 4.26, match those of a high-spin $Fe^{3+}$ (S = 5/2) center, consistent with a rhombic distortion E/D of about 0.22 and a positive D value, as observed for *Cp* Rd and sulfur ligated ferric iron model compounds[32,45]. Taken together, spectroscopic data demonstrate that both $Fe^{2+}$ and $Fe^{3+}$ are tightly bound into a tetrathiolate environment as in natural Rds, both in geometry and electronic structure.

Once established the high binding affinity of METPsc1 for iron in both oxidation states, we analyzed whether the protein accomplishes reversible redox cycles. We performed a typical redox-cycling experiment following changes of the characteristic $Fe^{3+}$METPsc1 band at 494 nm. We cyclically oxidized iron upon exposure to air, followed by argon purge and reduction by sodium dithionite addition (Fig. 5a).

A FeMETPsc1 solution (40 μM, pH 7) was subjected to at least twelve consecutive and reversible redox cycles, without any loss of the protein signal upon recycling (Fig. 5b), similarly to other redox-cycling Rd mimics[30–32]. The cycling experiment lasted two days, and the complex was kept under argon atmosphere overnight without any detectable loss of signal and full recycling for two more times the day after (see Supplementary Fig. 8). The last of 12 oxidation processes recovered approximately 92% of the expected $Fe^{3+}$METPsc1 signal, suggesting that more cycles could be performed. These results demonstrate that FeMETPsc1 can reversibly switch between ferrous and ferric states in diffusion under excess of reductant (dithionite) or oxidant (dioxygen), respectively.

Next, we performed electrochemical measurements to assess how the mutation of Tyr11 and Val44 in *Cp* Rd with two Ala residues (Ala7 and Ala22 in METPsc1, respectively) would affect the redox potential, and confirm the correlation to the designed structure. Indeed, a double mutant in such positions has never been reported in *Cp* Rd to date, and thus it was of particular interest to analyze METPsc1 electrochemistry. We carried out cyclic voltammetry experiments at different scan rates in which a glassy carbon electrode was immersed in a solution of 80 μM FeMETPsc1 (pH 7), using 0.3 M KCl as electrolyte (Fig. 6a).

FeMETPsc1 gave measurable currents in the range of 2.5–50 mV/s, displaying a quasi-reversible behavior with reduction potential centered at $E'^0 = 121$ mV (*vs* SHE), with $\Delta E_p$ in the range 59–136 mV. This high potential was our design goal, and it is not surprising considering the crystallographic data. The number and strength of H-bonds in the second coordination sphere (Ala7, Ala22, Asn19, Arg27) significantly decrease the electron density of sulfur donors, thus favoring the ferrous state. Randles-Ševčík analysis has been used to evaluate the diffusion coefficients of the reduced and oxidized species (Fig. 6b). They are $0.92 \cdot 10^{-6}$ and $1.4 \cdot 10^{-6}$ cm² s⁻¹ for the reduced and oxidized forms, respectively, in reasonable agreement with the value calculated from the crystallographic model ($1.47 \cdot 10^{-6}$ cm² s⁻¹).

### Definition of an artificial photo-triggered electron cascade

FeMETPsc1 possesses a high reduction potential value (121 mV *vs* SHE), slightly higher than the values observed for prokaryotic Rds (−100/+50 mV)[19]. The $Fe^{3+}$ reduction to $Fe^{2+}$ is accompanied by a clear change in the visible spectrum (see Fig. 5a), as expected. These results altogether prompted us to design a photo-triggered reduction experiment, to test whether FeMETPsc1 could represent the final electron acceptor of an electron transport chain, (Fig. 7a). To this end, we used an artificial porphyrin-containing miniature protein as the light-harvesting unit. In particular, triethylamine (TEA) was chosen as sacrificial reductant, and FeMETPsc1 as oxidant, whilst the newly

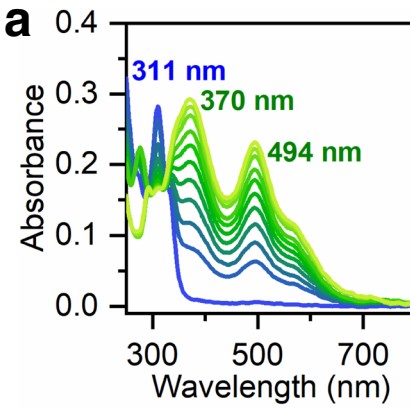
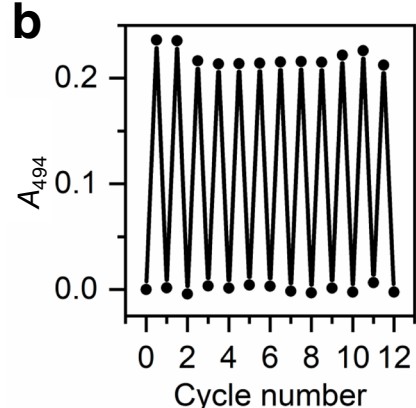

**Fig. 5 | FeMETPsc1 redox characterization. a** UV-Vis monitoring of Fe$^{2+}$METPsc1 (blue trace) aerobic oxidation to Fe$^{3+}$METPsc1 (lime trace). Spectra were acquired every 3 min. 40 µM FeMETPsc1, 20 mM HEPES buffer 2 mM TCEP, pH 7. **b** Redox cycling of FeMETPsc1 (40 µM) in HEPES buffer (20 mM) and TCEP 2 mM, pH 7, monitored by absorption at 494 nm (corresponding to the ferric species). Cycles consist of successive (i) air purge of the Fe$^{2+}$ complex to form the Fe$^{3+}$ complex and (ii) argon purge and dithionite reduction to restore the Fe$^{2+}$ complex. Source data are provided in a Zenodo repository under accession code 7748883.

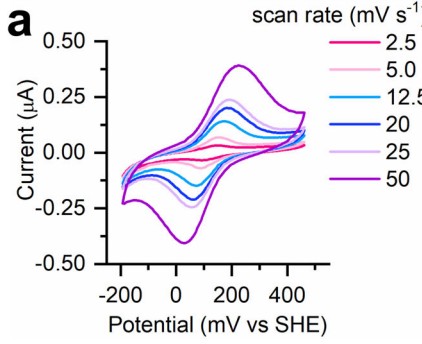
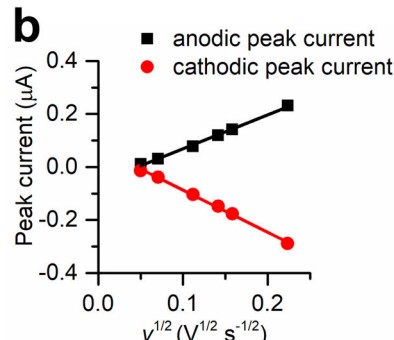

**Fig. 6 | FeMETPsc1 electrochemical characterization. a** Cyclic voltammograms of FeMETPsc1 (80 µM) recorded at different scanning rate from 2.5 to 50 mV s$^{-1}$ (bright pink to violet), in 40 mM HEPES buffer (pH 7) and 0.3 M KCl. Each voltammogram is the last of three consecutive scans. **b** Anodic and cathodic peak currents derived from cyclic voltammetry experiments of FeMETPsc1 (80 µM) plotted as a function of the square root of the scan rate. Data points were fitted to the Randles-Ševčík equation, allowing to determine the diffusion coefficient of FeMETPsc1 both in the reduced ($D$red = 0.92 10$^{-6}$ cm$^2$ s$^{-1}$) and in the oxidized state ($D$ox = 1.4 10$^{-6}$ cm$^2$ s$^{-1}$). Source data are provided in a Zenodo repository under accession code 7748883.

synthesized Zn$^{2+}$ derivative of MC6*a (ZnMC6*a) was used as photosensitizer[12]. Zinc tetrapyrroles have been already used in designed and engineered metalloproteins, and they showed peculiar time-resolved spectroscopic features[46], intra-molecular ET processes[5,6,10], and allosteric modulation[47]. However, this photoactive cofactor has never been used to transfer electrons from one protein to another. Therefore, a simple experiment was carried out by following FeMETPsc1 UV/Vis-spectrum differences upon reduction/oxidation due to green light exposition (Fig. 7b).

When a solution containing 2 mM TEA, 40 µM Fe$^{2+}$METPsc1, and 5 µM ZnMC6*a was purged with air, a 494 nm band of the oxidized [FeCys$_4$]$^{1-}$ appeared (Fig. 7c, d), demonstrating that iron oxidation at METPsc1 was not affected by TEA and ZnMC6*a. When the solution was exposed to green light irradiation for 20 min under argon atmosphere, almost complete disappearance of the ferric charge-transfer band was observed. A band at 311 nm concomitantly appeared, characteristic of the reduced [FeS$_4$]$^{2-}$ species (Fig. 7c). As a control, when the system was kept under Ar atmosphere in the dark for 30 min, the signal at 494 nm decreased only of approximately 10% (Fig. 7d, blue box). These results clearly demonstrate Fe$^{3+}$METPsc1 reduction upon light exposure. As a final proof of the artificial photo-electron transfer chain, the system was exposed to air and then to green light irradiation for three times. As expected, air oxidized Fe$^{2+}$METPsc1, and then after 20 min of irradiation, it was reduced back with formation of a peak at 311 nm. However, only partial disappearance of the band in the visible region could be observed in the following cycles, with a Fe$^{2+}$METPsc1 signal corresponding to almost half of the oxidized species (from 35 µM to 15 µM of Fe$^{3+}$METPsc1 concentration). Incomplete reduction was indeed accompanied by ZnMC6*a degradation after each cycle (see Supplementary Fig. 9a). In turn, this could be ascribed either to reactive oxygen species that formed during the previous O$_2$ reduction step by FeMETPsc1 (Fig. 7a), or by formation of radical species due to self-oxidation. ZnMC6*a was therefore exposed to 20 min irradiation in the absence of FeMETPsc1. Notably, in only one irradiation round, ZnMC6*a was fully converted to degradation byproducts, lacking the characteristic Soret band (Supplementary Fig. 9b).

## Discussion
The first goal of this work was to convey a general methodology for the design of miniature redox proteins and secondarily to demonstrate its applicability by developing a fully artificial electron transport chain. To do so, we constructed a high-potential miniprotein, leveraging from the wealth of mutagenesis studies on Rds. Rd from *Clostridium pasteurianum* (*Cp*) has been a central player in unraveling the factors that affect the reduction potential of the FeCys$_4$ metal site[19,20]. It was shown that Fe(II) stability can be related to the number and the strength of H-bonds involving the coordinating Sγ atoms, and may require

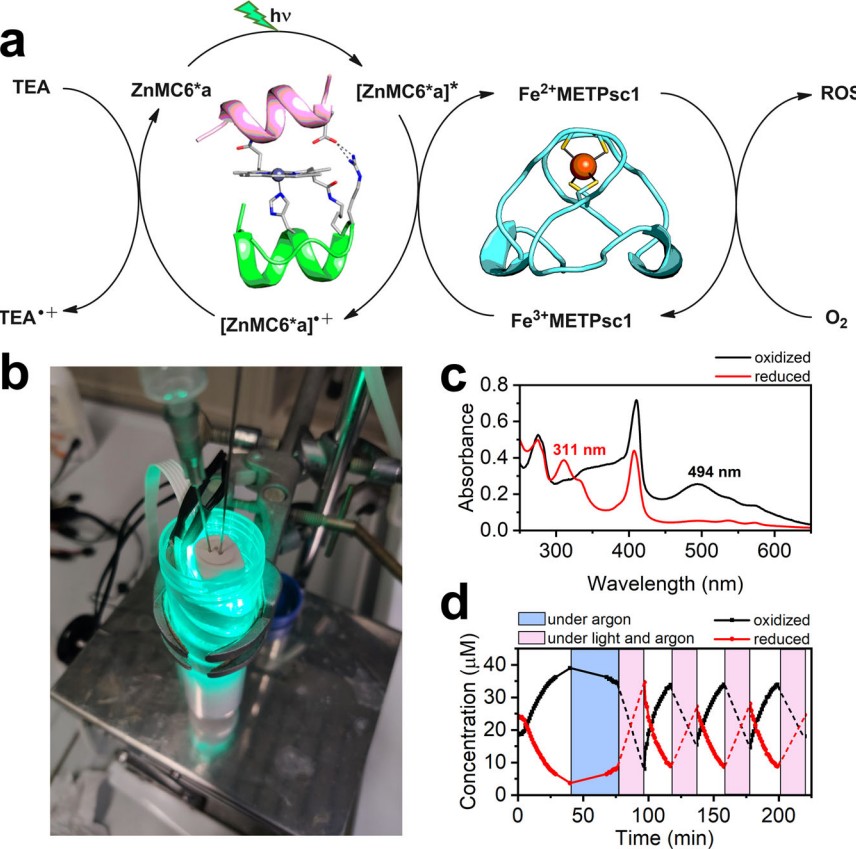

**Fig. 7 | Photoinduced electron transfer from ZnMC6\*a (5 μM) to Fe³⁺METPsc1 (40 μM). a** Reaction scheme of the synthetic electron cascade. **b** experimental setup showing the LED strip wrapped around the UV cuvette under Ar atmosphere. **c** superimposed UV-Vis spectra of Fe³⁺METPsc1 (black trace) and Fe²⁺METPsc1 (red trace) in the presence of ZnMC6\*a (5 μM) and triethylamine (2 mM). **d** redox cycling of FeMETPsc1 monitored at 496 nm and 311 nm (molar absorptivities are reported in Table 1). Pink boxes correspond to 20 min of green light irradiation, blue box corresponds to the dark control under Ar atmosphere. Source data are provided in a Zenodo repository under accession code 7748883.

multiple mutation while preserving the global fold and the expression profile[25,26].

The combination of powerful computational tools[48,49], and more recently machine learning[50,51], together with the genome palette (e.g., directed evolution and phage/yeast display)[52,53] is significantly helping protein designers in increasing success rate, adapting the protein to host the desired mutations. However, direct correlation between single point mutations and metal-dependent function still remains elusive when large scaffolds are adopted[52,54]. De novo design provides a remarkable alternative, which allows incorporating all the desired mutations at once, and generating the most suited structural arrangement for testing and refining folding and functions, such as redox potential[55]. Design of synthetic metalloproteins by miniaturization is particularly helpful, limiting the metal surroundings to only a few crucial residues[56,57]. Therefore, it was of considerable interest to develop by design and miniaturization a synthetic Rd, METPsc1, and show that it is capable of keeping the intended structural and functional properties in a small 28-residue peptide. Our design strategy, differently from previous attempts mimicking Rd[31,32], is fully generalizable, because it relies only on the knowledge of the mutual orientation of the C₂-related moieties, without using neither a specific supersecondary motif nor cyclization/stapling to link them.

A fundamental test of the correctness of our design came from electrochemical measurements of the iron derivative. The high value of the reduction potential, which surpasses the classical range for prokaryotic Rds, and closely matches the potential of rubrerythrins[19,26] was our design goal, and well fit with the crystallographic data. In fact, the agreement between the high-resolution crystal structure and the designed model at sub-Å level not only validate the adopted design principles, but, most interestingly, proved that the designed second-shell interactions are crucial in determining one of the highest potentials amongst the Rd family. This result prompted us to generate a synthetic electron transfer chain from a sacrificial electron donor (TEA) to a sacrificial acceptor (O₂) by means of two newly-developed synthetic mini-proteins (FeMETPsc1, ZnMC6\*a), whose overall size correspond to ~6.5 kDa.

In perspective, our studies provide a prototype for the generation of nanosized multicomponent mini-protein devices. They should encourage future design of small metalloproteins with predetermined structural and functional properties.

## Methods

Computational modelling and simulation methodology is described in the Supplementary Information.

### Solid-phase peptide synthesis

METPsc1 was synthesized by automatic solid-phase synthesis, using an ABI 433 A peptide synthesizer (Applied Biosystem, Foster City, CA, USA) with standard Fmoc chemistry on a 0.1 mmol scale. The acid labile H-PAL ChemMatrix resin, with a substitution of 0.20 mmol/g, was used as solid support. Amino acids were activated in situ with 2-(7-Aza-1H-benzotriazole-1-yl)−1,1,3,3-tetramethyluronium hexafluorophosphate (HATU) as coupling reagent. The N-terminal amino group was acetylated with a solution of acetic anhydride, 1-hydroxybenzotriazole (HOBt) and diisopropylethylamine (DIEA) in N-methyl-pyrrolidone (NMP). Peptide cleavage from the resin and

sidechains deprotection was achieved with a mixture of trifluoroacetic acid/$H_2O$/triisopropylsilane/ethanedithiol 9.4:0.25:0.25:0.1 (v/v/v/v), yielding to amidated C-terminal. The crude peptide was precipitated in cold diethyl ether and dried under reduced pressure. The isolated crude product was obtained in 65% yield (based on the resin substitution), with 50% HPLC purity.

## Peptide purification and analysis

Peptide purification was accomplished using a Shimadzu LC-8A preparative HPLC system (Shimadzu, Kyoto, Japan), equipped with a SPD-M10AV UV-Vis detector. A linear gradient of $H_2O$ 0.1% TFA (eluent A) and acetonitrile 0.1% TFA (eluent B), from 5 to 70% B over 50 min at a flow rate of 22 mL/min, eluted a Reverse Phase Vydac C18 column (250 cm × 22 mm; 10 μm).

Peptide purity and identity were assessed by RP-HPLC-MS analysis (Supplementary Fig. 10–12), using a Shimadzu LC-10ADvp equipped with an SPDM10Avp diode-array detector. ESI-MS spectra were recorded on a Shimadzu LC-MS-2010EV system with ESI interface and a quadrupole mass analyzer. A Vydac C18 column (150 mm × 4.6 mm, 5 μm) was used in the LC-MS analyses, eluted with a linear gradient of $H_2O$ 0.1% TFA (eluent A) and acetonitrile 0.1% TFA (eluent B), from 5 to 70% B over 60 min at a flowrate of 0.5 mL/min.

## Crystallography

The ZnMETPsc1 complex was crystallized by the hanging drop vapor diffusion method at 20 °C. Typically, a drop containing 2.0 μL of 1:1 (v/v) mixture of protein solution (10 mg/mL, 7 mM DTT, 4 mM $ZnCl_2$) and 2.0 μL of precipitant buffer (0.1 M HEPES at pH 7.5, 1.4 M sodium citrate tribasic dihydrate) was equilibrated against 0.5 mL reservoir of precipitant buffer. Crystals of the ZnMETPsc1 complex appeared within 4 days and grew as long needles with typical dimension of 0.15 × 0.15 × 0.5 mm³. Crystals were transferred to the same mother liquor solution augmented with 30% MPD solution and flash cooled. These crystals yielded diffraction data to 1.34 Å resolution at the XRD1 beamline (Elettra Synchrotron Light Source, Trieste, Italy), using a wavelength of 1.2400 Å, and kept at 100 K. Data were processed using XDS and POINTLESS (version 1.11.21)[58,59] with a data collection statistics reported in Supplementary Table 2. Crystals grew in the orthorhombic space group C2221. No twinning was detected.

The structure of the ZnMETPsc1 complex was solved by molecular replacement via Phaser[60], run under Phenix suite (version 1.16)[61], using the designed model cleaved of the N- and C-terminal residues as a search model. The optimal solution for the positioning of one monomer in the asymmetric unit yielded a total log-likelihood gain of 21, a rotation function Z score (RFZ) = 3.2 and a translational function Z score (TFZ) = 3.7. An initial rigid-body refinement with data at 2.5 Å dropped the R/Rfree to 0.377/0.427. The program PHENIX.refine was used to anisotropically refine the model, and the graphics program COOT[62] was used for structural model adjustments and inspection of Fourier residual maps. In the final stage of refinement, a total of 26 water molecules could be located. The data processing and structural refinement statistics are shown in Supplementary Table 2.

Protein Data Bank has been accessed (March 11, 2022) for high-resolution Rd structures in order to determine the average $M^{2+}$−Sγ distance[40]. The search settings were: "Uniprot Molecule Name" contains "Rubredoxin", "Refinement Resolution" >0.5 and < =1.2 Å. A total of 25 entries were retrieved. Among them, only 4 contained $Zn^{2+}$ as ligand, for a total of 12 independent models binding zinc in the $Cys_4$ binding site.

## UV-Vis Spectroscopy

UV-Vis spectra were acquired on a Cary Varian 60 spectrophotometer, equipped with a thermoregulated cell holder and a magnetic stirrer. All buffer, protein or metal solutions were prepared with MilliQ water and purged with argon. All experiments were performed at 25 °C, using rubber sealed quartz cuvettes of 1 cm pathlength. Concentration of METPsc1 was determined using a molar extinction coefficient of $\varepsilon_{280} = 2980$ $M^{-1}$ $cm^{-1}$. UV-Vis titration experiments with $Fe^{2+}$ were performed by adding aliquots (~0.1 equiv) of Mohr's salt to a solution of apo-METPsc1 (30 μM) in HEPES buffer (20 mM) pH 7 containing 1 mM TCEP. In the redox cycling experiment, a 0.7 mL solution of METPsc1 (50 μM) in HEPES buffer (20 mM) and TCEP (2 mM) at pH 7 was preliminary purged for 5 min with Ar and then a 10 mM Mohr's salt solution under Ar atmosphere was added to a final concentration of 40 μM. Next, the solution was sequentially purged with air to form the $Fe^{3+}$ complex, then with argon and finally reduced with 0.2 μL of 0.5 M sodium dithionite, prepared under Ar atmosphere, to restore the $Fe^{2+}$ complex. UV-Vis spectra were acquired every 3 min.

## Circular dichroism spectroscopy

CD spectra were recorded at 25 °C on a JASCO J-815 dicrograph equipped with a thermoregulated cell holder. All spectra were acquired at 0.2 nm intervals with 20 nm/min scan speed, using quartz cells of 1 cm pathlength. Spectra in the UV-visible region (300–800 nm) were collected for the oxidized and reduced forms of FeMETPsc (40 μM) in HEPES buffer (20 mM) at pH 7. The $Fe^{2+}$ complex was prepared by addition of Mohr's salt (1.5 equiv) to an argon purged solution of METPsc1. The latter was then purged with air to obtain the $Fe^{3+}$ complex.

## Electron paramagnetic resonance spectroscopy

For the EPR study, $Fe^{3+}$METPsc1, in 20 mM phosphate buffer (pH 7) and 5 mM TCEP, was mixed with 30% (v/v) glycerol to an approximate final concentration of 0.5 mM. A Bruker Elexys E580 X-band spectrometer (microwave frequency 9.76 GHz) equipped with a cylindrical dielectric cavity and a helium gas-flow cryostat from Oxford Inc was used to acquire the CW-EPR spectra. The spectrum was recorded at 4.5 K and a microwave power of 1 mW, a modulation amplitude of 0.7 mT and a modulation frequency of 100 KHz were used.

## Cyclic voltammetry

All cyclic voltammetry experiments were performed under argon, with a Potentiostat/Galvanostat μAUTOLAB Type III (Metrohm Autolab, Utrecht, The Netherlands) using a three-electrode cell for small volume samples (0.5–2 mL) purchased from BASi (West Lafayette, IN, USA). Temperature-controlled measurements were conducted using a thermo-cryostat R2 (Grant). For all measurements, a 3 mm-diameter glassy carbon electrode (GCE, BASi) was used as working electrode. A Pt wire and an Ag|AgCl NaCl 3 M electrodes (BASi) were used as counter and reference electrode (E°' = 0.206 V), respectively. Acquired data was processed by GPES software package (v4.9).

Cyclic voltammetry experiments on freely diffusing FeMETPsc1 were performed at 15 °C, by adapting a previously published procedure[63]. A 5 μL drop of a 0.76 mM METPsc1 solution in water was deposited on a square piece of a Spectra/Por (Biotech CE MWCO 0.5–1 kDa), and 0.2 μL of a 100 mM Mohr's salt solution were added to it. Then, the polished GCE was pressed against the membrane and an O-ring, to form a solution layer. The electrode was then immersed in 20 mM HEPES buffer and 0.3 M KCl at pH 7 for 5 min to reconstitute the protein. The sample volume in the electrochemical cell was 2.0 ml. CV measurements were performed three times in the range 2.5–50 mV/s of scan speed, and the third voltammogram was used to perform the analysis. Diffusion coefficient of the crystallographic model was calculated by HYDRONMR[64].

## Photo-induced electron transfer

ZnMC6*a was synthesized according to previously described procedures[65]. A solution of $Fe^{2+}$METPsc1 (50 μM), ZnMC6*a (40 μM) and triethylamine (4 mM) in HEPES buffer (20 mM) pH 7 was prepared

and placed in a rubber sealed UV-Vis cuvette. The solution was first purged with air to form the Fe$^{3+}$METPsc1 complex, then purged with argon for 30 min prior to the photoreduction. The latter was achieved by wrapping the cuvette with a green led strip ($\lambda_{max}$ 570 nm, 5 mW/cm$^2$ per led bulb) for 20 min during each cycle, to keep a constant light dose.

## Reporting summary

Further information on research design is available in the Nature Portfolio Reporting Summary linked to this article.

## Data availability

The crystal structure of ZnMETPsc1 complex has been deposited in wwPDB with the accession code 5SBG[66]. The experimental and computational source data used in this study are available in the Supplementary Information and in the Zenodo database under accession code 7748883[67].

## Code availability

The custom codes used are available in the supplementary information file and in a Zenodo repository under accession code 7748883[67].

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

## Acknowledgements

We wish to thank Elettra Sincrotone Trieste for providing access to its syncroton radiation facilities, and Dr. Maurizio Polentarutti for X-ray data collection, Dr. Artemis Papadaki and Dr. Carlo De Luca for performing preliminary designability analysis and acquiring spectroscopic data, Prof. Flavia Nastri and Ornella Maglio for fruitful discussion and Dr. Monica Grasso for administrative support.
This work was supported by Campania Region "Programma Operativo FESR Campania 2014-2020, Asse 1" [CUP B63D18000350007] and by Italian MUR, Project SEA-WAVE 2020BKK3W9, [CUP E69J22001140005]. The authors also acknowledge the Italian MUR program "Dipartimenti di Eccellenza 2023-2027" for the projects arCHIMede [CUP E63C22003710006] (M.Chino, L.L., S.L.G., A.L., V.P.), and CH4.0 [CUP D13C22003520001] (A.F., M.Chiesa). We thank the Department of Chemical Sciences of the University of Naples Federico II for covering part of the publication fee. A.F. and M.Chiesa gratefully acknowledge financial support from the European Union's Horizon 2020 research and innovation program under the Marie Skłodowska-Curie Grant agreement no. 813209 (PARACAT).

## Author contributions

M.Chino and V.P. conceived the project and designed the miniproteins, which S.L.G. and L.L. synthesized and purified. M.Chino and L.L. performed the spectroscopic characterization and the electrochemical experiments; L.F.D.C., L.L. and S.L.G. conducted the crystallization and L.F.D.C. acquired crystallographic data; L.F.D.C. and M.Chino determined the X-ray crystal structure; M.Chiesa and A.F. acquired and analyzed EPR data; M.Chino and L.F.D.C prepared the manuscript draft; M.Chino, V.P. and A.L. interpreted the data, edited and finalized the manuscript with input from all authors; V.P. and A.L. supervised the project.

## Competing interests

The authors declare no competing interests.
