## [Peer Review File · Nature Communications]

Designed Rubredoxin miniature in a fully artificial electron chain triggered by visible lightREVIEWER COMMENTS

Reviewer #1 (Remarks to the Author):

The manuscript by Lombardi and co-workers describes a de novo design of a miniature protein with only 28 amino acid residues that mimics the function of rubredoxin in electron transfer. While it's indeed impressive that the X-ray structure of the artificial protein (with Zn as the metal center) can be resolved at such a high resolution, the current work suffers from several major drawbacks:

1) If the aim is to provide a general methodology for the de novo design of miniature proteins, the authors should have compared the differences and/or advantages of the current method with literature methods (especially with ref 28 and ref 29); if the aim is to obtain a fully functional miniature protein capable of transferring electron, now that the X-ray structure is available, isn't it the best opportunity to go a step further and do some protein engineering, especially the second coordination sphere residues, to obtain a more robust miniature protein? As this would not only test the authors' claim that "the designed second-shell interactions are crucial in determining one of the highest potentials amongst the Rd family.", but more importantly, could even higher redox potential possible via mutagenesis?

2) Redox cycling ability is an important feature for any de novo designed rubredoxin mimics, although the authors claim that "no dramatic loss of the protein signal upon recycling for 9 times", but as far as this reviewer can see from figure 4b, the Abs at 494 nm decreased very obviously, what is the possible reason? Would it be possible that the Fe-S center is not so stable during oxidation or dithionite-treatment? Once the Fe ion is lost in the process, what would happen to the Cys residues? This issue is also true for the experiment in Figure 6, as obvious decrease of the Abs at 314 nm for the reduced species can be observed only after the second redox cycling, what if a third of even fourth cycle? Is the protein robust/stable enough in this system?

As such, this reviewer does not think the manuscript fulfill the high publication standards of Nature Communications.

Below are some other minor issues that may be helpful for the author to revise the manuscript for another journal.

1) Why 5 mM of TCEP was included in the EPR sample?

2) In the experiment described in Figure 6, why the second round of light irradiation is only 15 min while the first one is 25 min? And by the way, it looks like only 10 min of the light irradiation was applied in the second round from Figure 6d, is it a drawing mistake?

3) The X-ray structure is resolved only for the Zn-complex, did the author try the Fe-complex instead?

4) In the SI, page 11, Supplementary Fig. 2 in the text should be Fig. 3; SI, page 13, Supplementary Fig. 3 in the text should be Fig. 4.

5) The yield of the peptide synthesis is 65%, is this HPLC yield or isolated yield? ESI-MS spectrum should be given for the purified peptide, rather than the crude, as multiple masses can be observed in the Supplementary Fig. 11.

Reviewer #2 (Remarks to the Author):

Overall, I found the described work to be exciting and relevant to the times. The authors created a novel nano-sized biological photoelectron acceptor that can be used as a tool for broader applications that require electron transport. The designed system contains a single polypeptide chain consisting of

only 28 residues that contains internal 2-fold symmetry. They effectively created a protein knot to tetrahedrally coordinate metal that is more structurally stable than a Zn-finger. They performed the appropriate biophysical experiments (EPR, mass spectrometry, UV-VIs titrations and circular dichroism) to prove the designed protein coordinated iron such that it could be reduced and oxidized on demand. Then crystallized the the protein with Zn²⁺ Ultimately, I would prefer the crystal structure contain iron instead of zinc, but I recommend the manuscript be published without any major changes. Below is a list of minor changes that should be made.

To improve the paper, the authors should have a figure of the Cp Rd with its first and second coordination spheres defined. They draw many comparisons to this structure throughout the manuscript and it would be easier for the reader to have a picture instead of having to pull up the structure from the PDB.

I found the crystal packing description on page 8 (last paragraph of section “ZnMETPsc1 crystal structure reveals a handful of secondary motifs”) to be a distraction and should be deleted. The description doesn’t lead to any relevant conclusions. I believe it was to setup the placement of the symmetry related Arg in the second coordination sphere- but is not needed. Also Supplemental, Figure 8: This figure doesn’t add any special information to the paper and can be removed. The figure is too busy to full understand with out being able to rotate it. The placement of the Asp and Arg residues is not obvious. I assume the red balls signify water molecules.^[SEP]
Figure 2, part d: add the distances for the coordinating bonds.

The rest of the requested changes are typos and bad sentence structure that can easily be corrected.

Page 2: lines 6-9; re write: “We assembled in a as small as 28-^[SEP] residue peptide the quintessential elements required to correctly fold around a single iron redox center, coordinated to four cysteinyl thiolates (FeCys4 site), and to efficiently function in electron- transfer.”^[SEP]

Page 3: line 14 : put a comma after “ to achieve for the first time in protein design”

Page 3: line 16: Nature should not be capitalized.

Page3: line 23: clarify the statement about rubrerythrin.

Page 4: line 16: restate “ half-sized respect to natural Ads”

Page 4: line 25: “second sphere to second coordination sphere

Page 5: line 1 : Sulpur should not be capitalized.

Page 5, line 25: change “identifying” to “identify”

Page 6, line 4: packer??

Figure 1: legend, line 10: change “found with” to “found from”

Page 7, line 8 & page 9, line 6: change “buldge “ to “bulge”

Page 7, lines 7, 8 & 9: change “2-residues” to “2-residue”

Page 7, line 18: Sentence starting with The overall backbone should be rewritten.

Page8, lines 11 & 12: Why is this sentence important ? ”In addition, interactions between Tyr16 and the equivalent residue of a crystallographic related METPsc1 molecule are observed with a 3.32 Å distance between -OH atom groups. “

Page 8: the crystal packing information is not relevant.^[SEP]

Page 8, lines 20 & 21: change “and containing Zn²⁺” to “that contain Zn²⁺”

Page 10, line 18: change “(one and two order of magnitude, respect to Zn²⁺ and Co²⁺, respectively)” to “(one and two orders of magnitude for Zn²⁺ and Co²⁺, respectively)”

Page 10, line 19: change “most probably attributable to “ to “most likely attributable to “^[SEP]

Page 12, Figure 4: define the Fe²⁺ and Fe³⁺ species in part b.^[SEP]

^[SEP]

Page 19: line 5: “statistic” to “statistics”

Page 19 lines 5 & 6: “Crystals presented an orthorhombic unit cell with space group C2221. “ change

to something like “ Crystals grew in the orthorhombic space group C2221.”

Page 19: line 25 & Page 20, line 10: change “25°C” to “25 °C”

Page 20: line 18: Change “30% of glycerol as glassing agent to an” to “30% glycerol to an”

Page 21: line 2: move “ under argon” to “All cyclic voltammetry experiments were performed under argon with a Potentiostat...”

Page 21, line 6: change “For all the measurement” to F”or all measurements”^[SEP]Page 21: lines 10 & 11: clarify “Cyclic voltammetry experiments on freely diffusing FeMETPsc1 were performed by adapting a previously published procedure, at 15 °C”

Page 22, lines 1 & 6: availability misspelled.

Authors need to define acronyms at the first instance of use:

some examples: Page 3: line 23: define SHE before using acronym

Page 4; line 7: define METP

Page 5; line 11: define Cp

Wavelength of data collection differs between main manuscript (1.00 Å) and supplementary table 2 (1.24 Å).

Resolution of refined structure also differs between main manuscript (1.44 Å) and supplementary table 2 (1.34 Å).

Items missing from supplementary table 2: redundancy of data, add the RSCC and RSR of Zn ions, add the mol probity score and clashscore, footnotes explaining the terms.

Supplemental, page 6: Define where to find the MASTER software as you did PyMOL.

Supplemental, page 7: change “grigoryanlab” to “Grigoryan lab “

Supplemental, Page 11: (Supplementary Fig. 2) should be (Supplementary Fig. 3)

Supplemental, page 13 (Supplementary Fig. 3) should be (Supplementary Fig. 4)

Supplemental, pages 15 & 16: Change” The C-terminal strand (Strand F) folds in a head-to-tail manner to antiparallelly align with the first strand, “ to “ The C-terminal strand (Strand F) folds antiparallel with respect to the first strand,”

Reviewer #3 (Remarks to the Author):

Designed Rubredoxin miniature in a fully artificial electron chain triggered by visible light by the group of Lombardi and Pavone

The group has important and valid contributions in designing metal sites into de novo proteins, and in engineering of metalloprotein functions in designed and native scaffolds, in the field of artificial enzymes (heme proteins).

In this manuscript, the authors assembled a small peptide required to correctly fold around a single iron redox center, coordinated to four cysteinyl thiolates (FeCys4 site), a mimetic compound for Rubredoxins that can participate and function in electron-transfer.

(Fe³⁺, Fe²⁺) METPsc1 and Zn²⁺+METPsc1 were synthesized with success and the crystal structure Zn²⁺+METPsc1 was detailed analysed. Fe³⁺+METPsc1 was explored by spectroscopic methods (VIS, CD, EPR) and electrochemical properties determined, always with counterpoint with the ferrous form. The high reduction potential compared to natural and designed FeCys4-containing proteins was exploited as a terminal electron acceptor of a fully artificial chain triggered by visible light.

The work is exciting and deserves to be publish.

A few points to clarify:

i) The METPsc1 ligand was obtained in large amounts. Fe and Zn could be reconstituted in the template. This is a fact well known for rubredoxins that can also accept a large number of metals (mainly transition metal) and even Ga can be used as an interesting isomorphous replacement, for Fe³⁺. Why the crystal structure was extensively done in for and Zn²⁺+METPsc1? I understand the spectroscopy was conducted for (Fe³⁺, Fe²⁺) METPsc, since Zn²⁺ is silent in EPR and no absorption bands are observed in Vis region (as well no redox).

ii) Table I compares the spectroscopic parameters of FeMETPsc1 and Cp Rd in Fe(II) and Fe(III) oxidation states. There is a clear match.

The EPR data for Fe²⁺ states should not be indicated as (-) is a S=2 state, as it looks EPR silent (difficult to be observed by EPR using different modes, and really observed by MB techniques). High-spin integer spin Fe²⁺ (S = 2) is more difficult to observe by EPR methods and low-spin Fe²⁺ is EPR silent. Both oxidation states in Rd type proteins are high-spin systems. Mössbauer (MB) spectroscopy is a particularly suitable technique for investigating the valence and spin-states of iron sites in Rd.

iii) Figure 4. FeMETPsc1 redox characterization. a, UV-Vis monitoring of Fe²⁺+METPsc1 (blue trace) aerobic oxidation to Fe³⁺+METPsc1 (red trace).
blue trace or purple trace ???

iv) The FeMETPsc1 possesses significantly high potential value (121 mV vs SHE), exceeding the typical range for prokaryotic Rds (-100/+50 mV). The phrase could be modified ... an high reduction potential value (121 mV vs SHE), slightly higher than the values observed for Rds.

v) Photoinduced electron transfer from ZnMC6* to Fe³⁺+METPsc1 is an interesting observation, as a possible reaction scheme of the synthetic electron cascade. Other couples could be explored. More details should be given on ZnMC6*.

Reviewer #4 (Remarks to the Author):

This manuscript describes the design and synthesis of a functional artificial rubredoxin (Rd)-like FeCys₄-cluster peptide. This miniaturised metallo-‘protein’ (METPsc1) was designed de novo by dissecting the Fe-binding region from a high-resolution structure of Rd and then searching for the optimal peptide fragment that could bridge the two truncated sequences. The final model was obtained by Monte Carlo sampling. The Rd miniature was synthesised and characterised by X-ray diffraction as the Zn complex. Coordination to the metal closely mirrors that of native Rd and matches exceptionally well onto the computationally modelled structure. Fe binding was then assessed using UV-Vis, CD and EPR; a 1:1 complex is formed with the metal and the cluster gives characteristic LMCT UV-Vis bands and EPR resonances. Reversible redox cycling is also demonstrated. CV of the cluster reveals a high reduction potential relative to native Rds. Finally, Fe-METPsc1 is used as the terminal electron acceptor in a visible-light-mediated artificial electron cascade.

Based on the fact that the Fe-METPsc1 was designed from ‘scratch’ and very well characterised, including X-ray data, the novelty/originality of this paper is high. The integration of the FeMETPsc1 into an electron cascade further enhances the impact of the publication and points towards potential applications for de novo designed proteins in complex artificial systems. The methods described in the main paper, and details in the SI, are sufficiently detailed to allow reproduction of the results.

The computational design aspect of this work is outside my area of expertise so I will leave it to other reviewers to comment on the method; however, due to the extensive characterisation of the resulting METPsc1, I have no doubt regarding the effectiveness of the approach. My only real concern is in regard to Figure 6c/6d; the UV-Vis LMCT band at 496 is visible only as a flat absorbance next to the large band at around 400 nm from ZnMC6*a. Furthermore, the shift from 311 nm to 314 nm is very slight. These results are complicated by the fact that the ZnMC6*a compound absorbs at 496 nm after

successive rounds of oxidation and reduction. Since the final results of this paper are based on the disappearance of the 496 nm absorbance and 311 to 314 nm shift, the authors should include the UV-Vis traces (overlaid if clear or with Abs values if not) for all time points of Figure 6d in the SI to show the gradual changes in each peak. It would be ideal if this experiment could be repeated at a higher concentration (if that is possible) to give a more pronounced band at 496, however, this is not necessary if the UV-Vis spectra of the time points from Figure 6d are clear. Finally, it's not clear to me how the concentration values for the oxidised and reduced clusters are calculated for Figure 6d. Please add this info into the SI.

A few additional points:

The METP abbreviation should be clearly defined on first use.

For Figure 3, the four plots a, b, c, d are positioned clockwise which isn't standard formatting (compared to c below a and d below b) – a very minor point.

The amount of dithionite added for each redox cycle (Figure 4b) should be stated in the SI – an 'excess' is mentioned in the text, but clarity would aid reproduction of the experiments.

Line 6-7 of the abstract (“... in a as small as...”) should be re-worded for clarity.

In my opinion, this manuscript is of significant novelty and impact for publication in Nature Communications, and should be accepted once the above concerns have been addressed.

We would like to thank the reviewers for their comments and suggestions that have contributed to improve the quality of the manuscript.

We have considered the comments of all Reviewers and accepted all relevant suggestions, also by performing new experiments. In particular, we carried out again the redox cycling characterization (Figure 4), the photoinduced electron transfer experiments (Figure 6), and we acquired new LCMS spectrum of the purified peptide (Supplementary Information).

A detailed point-to-point reply to the Reviewers' comments, highlighting the changes made in the revised manuscript, is provided in the following.

REVIEWER COMMENTS

AUTHOR RESPONSE

Reviewer #1 (Remarks to the Author):

The manuscript by Lombardi and co-workers describes a de novo design of a miniature protein with only 28 amino acid residues that mimics the function of rubredoxin in electron transfer. While it's indeed impressive that the X-ray structure of the artificial protein (with Zn as the metal center) can be resolved at such a high resolution, the current work suffers from several major drawbacks:

1) If the aim is to provide a general methodology for the de novo design of miniature proteins, the authors should have compared the differences and/or advantages of the current method with literature methods (especially with ref 28 and ref 29); if the aim is to obtain a fully functional miniature protein capable of transferring electron, now that the X-ray structure is available, isn't it the best opportunity to go a step further and do some protein engineering, especially the second coordination sphere residues, to obtain a more robust miniature protein? As this would not only test the authors' claim that "the designed second-shell interactions are crucial in determining one of the highest potentials amongst the Rd family.", but more importantly, could even higher redox potential possible via mutagenesis?

We thank the Reviewer for pointing this out, giving us the opportunity to clarify our aim and put our work under a better focus. When writing this manuscript, we were primarily motivated to report the remarkable high-resolution structure of an artificial metalloprotein (as the Reviewer has acknowledged), and the impact of the designed and experimentally found interactions in determining a reduction potential that closely matches the highest reported potential in the Rd family. We feel that these are notable results of interest to the chemistry community. Indeed, they allowed us to develop a fully artificial electron transport chain.

Thanks to the Reviewer's comments, we have made changes in the text, to make these points clearer.

- 1. We have better highlighted the advantages of the adopted design methodology on p. 7 l. 5-7: "Interestingly, this search allowed us to overcome some of the limitations previously encountered to covalently link the two symmetry-related moieties, such as the use of stabilizing long β -hairpins²⁸ or synthetically difficult cyclization steps²⁹." and on p. 18 l. 12-15: "Our design strategy, differently from previous attempts mimicking Rd^{28,29}, is fully generalizable, because it relies only on the knowledge of the mutual orientation of the C₂-related moieties, without using neither a specific super-secondary motif nor cyclization/stapling to link them."*
- 2. Moreover, thanks to the new experiments we performed, we have been able to demonstrate the high stability of our miniaturized de novo metalloprotein in redox cycling, see p. 13-14 l. 17-20, 1-3, and Supplementary Figure 8: "A FeMETPsc1 solution (40 μ M, pH 7) was subjected to at least twelve consecutive and reversible redox cycles, without any loss of the protein signal upon recycling (Fig. 4b), similarly to other redox-cycling Rd mimics²⁷⁻²⁹. The cycling experiment lasted two days, and the complex was kept under argon atmosphere overnight without any detectable loss of signal and full recycling for two more times the day after (see Supplementary Fig. 8). The last of 12 oxidation*

processes recovered approximately 92% of the expected Fe³⁺METPsc1 signal, suggesting that more cycles could be performed.

We are confident that in the next future we and others could apply our simple approach to the design of many other miniaturized metalloproteins. Nonetheless, we agree with the Reviewer about the opportunity to perform a mutagenesis study to modulate the redox potential (such study is currently under course in our lab). However, we respectfully think that the electrochemical study of a conspicuous set of mutants (not only in the second but also in the first coordination sphere, possibly with non-coded amino acids) would deserve a full and more detailed paper, in which we would hopefully be able to expand the reduction potential range of the FeS₄ metal site. We are afraid that by conveying another different “story” in this paper would be misleading for the reader. We hope that the Reviewer could recognize that this manuscript already addresses important challenges.

2) Redox cycling ability is an important feature for any de novo designed rubredoxin mimics, although the authors claim that “no dramatic loss of the protein signal upon recycling for 9 times”, but as far as this reviewer can see from figure 4b, the Abs at 494 nm decreased very obviously, what is the possible reason? Would it be possible that the Fe-S center is not so stable during oxidation or dithionite-treatment? Once the Fe ion is lost in the process, what would happen to the Cys residues? This issue is also true for the experiment in Figure 6, as obvious decrease of the Abs at 314 nm for the reduced species can be observed only after the second redox cycling, what if a third of even fourth cycle? Is the protein robust/stable enough in this system?

*We thoroughly appreciate this comment, which allowed us to deepen and clarify this point. In the experiment reported in the original version of the manuscript, we did not allow the complete oxidation of the Fe-S center, because we used a fixed time for each oxidation cycle. Now, we re-ran the redox cycle experiment several times, allowing each cycle the complete oxidation to the ferric complex. We ran a total of 12 cycles with no apparent iron loss. The 10th cycle was performed overnight, and the next day we performed two more redox cycles to verify the complex stability over a long-time frame. In the revised manuscript, we provide the experimental data as a new Figure 5b and the full time-course with the overnight specification in Supplementary Figure 8. Remarkably, a huge beneficial effect was firstly observed by working with 0.8 eq of iron, thus excluding any effect of free iron to the redox cycling (most probably by Fenton chemistry), and secondly by using a dithionite solution, stored under argon atmosphere (most probably by preventing the formation of sulfides and sulfites by disproportion, which caused iron precipitation after a few additions, and/or sulfonation of the cysteines). We have added these experimental details on p. 21, l. 19-24: **“In the redox cycling experiment, a 0.7 mL solution of METPsc1 (50 μM) in HEPES buffer (20 mM) and TCEP (2 mM) at pH 7 was preliminary purged for 5 min with Ar and then a 10 mM Mohr’s salt solution under Ar atmosphere was added to a final concentration of 40 μM. Next, the solution was sequentially purged with air to form the Fe³⁺ complex, then with argon and finally reduced with 0.2 μL of 0.5 M sodium dithionite, prepared under Ar atmosphere, to restore the Fe²⁺ complex”.***

*We also recollected data for the light-induced electron transfer experiment in Figure 7, under slightly different experimental conditions (as suggested also by Reviewer #4). In the new experiment, we limited and fixed the irradiation time to 20 minutes for each cycle, to keep the light dose constant over each cycle. We clearly show that FeMETPsc1 is able to perform several redox cycles under these conditions. Moreover, it should be noted that the instability of the ZnMC6*a upon several irradiations is limiting the complete recycling.*

We hope that these new experiments have fulfilled the main concerns of the Reviewer, who might positively reconsider our revised version.

Minor issues:

1) Why 5 mM of TCEP was included in the EPR sample?

The Tris(2-carboxyethyl)phosphine (TCEP) is a common reducing agent used in peptide/protein-containing samples. We used it to ensure full reduction of thiol moieties before metal addition. We generally kept a TCEP:METPsc1 ratio in the range 20-30 (TCEP: 1 mM; METPsc1 30-50 μM). In the case of the EPR experiment a higher peptide concentration was needed (0.5 mM). Thus, we used a lower TCEP:METPsc1 ratio (= 10), considering that the sample was freshly prepared and immediately frozen under liquid nitrogen for data acquisition.

2) In the experiment described in Figure 6, why the second round of light irradiation is only 15 min while the first one is 25 min? And by the way, it looks like only 10 min of the light irradiation was applied in the second round from Figure 6d, is it a drawing mistake?

*We appreciate the comment by the Reviewer as it gave us the opportunity to significantly improve this point in the manuscript. It was not a drawing mistake: under the used experimental conditions (ZnMC6*a: 40 μ M; Fe³⁺METPsc1: 50 μ M), approximately 25 minutes were sufficient to reduce almost 100% of the FeMETPsc1. In the second round, we stopped the irradiation after 10 minutes because the absorption spectrum of the Zn-porphyrin moiety began to change, indicating porphyrin bleaching, and the Fe³⁺METPsc1 reduction was not proceeding any further. In the data we re-collected, we performed the experiment under a different ratio of FeMETPsc:ZnMC6*a of 40 μ M:5 μ M, to better highlight and evaluate the redox cycles of the Rd mimic. Moreover, we kept fixed the irradiation time at 20 minutes. Under these conditions, up to four cycles could be performed, even though 100% reduction was not obtained. Nevertheless, we hope that the Reviewer agrees with us that this result is quite valuable, also considering a possible degradation of ZnMC6*a. We believe that our system represents a stimulating proof of principle, even though it deserves further improvements.*

*The text was modified on p. 16/17, l. 9-14, 1-14, accordingly to the new findings: “When a solution containing 2 mM TEA, 40 μ M Fe²⁺METPsc1, and 5 μ M ZnMC6*a was purged with air, a 494 nm band of the oxidized [FeCys4]1- appeared (Fig. 6c,d), demonstrating that iron oxidation at METPsc1 was not affected by TEA and ZnMC6*a. When the solution was exposed to green light irradiation for 20 minutes under argon atmosphere, almost complete disappearance of the ferric charge-transfer band was observed. A band at 311 nm concomitantly appeared, characteristic of the reduced [FeS₄]2- species (Fig. 6c). As a control, when the system was kept under Ar atmosphere in the dark for 30 minutes, the signal at 494 nm slightly decreased (approximately 10%; Figure 7d, blue box). These results clearly demonstrate Fe³⁺METPsc1 reduction upon light exposure. As a final proof of the artificial photo-electron transfer chain, the system was exposed to air and then to green light irradiation for three times. As expected, air oxidized Fe²⁺METPsc1, and then after 20 minutes of irradiation, it was reduced back with formation of a peak at 311 nm. However, only partial disappearance of the band in the visible region could be observed in the following cycles, with a Fe²⁺METPsc1 signal corresponding to almost half of the oxidized species (from 35 mM to 15 mM of Fe³⁺METPsc1 concentration). Incomplete reduction was indeed accompanied by ZnMC6*a degradation after each cycle (see Supplementary Fig. 9a). In turn, this could be ascribed either to reactive oxygen species that formed during the previous O₂ reduction step by FeMETPsc1 (Fig. 7a), or by formation of radical species due to self-oxidation. ZnMC6*a was therefore exposed to 20 minutes irradiation in the absence of FeMETPsc1. Notably, in only one irradiation round, ZnMC6*a was fully converted to degradation byproducts, lacking the characteristic Soret band (Supplementary Fig. 9b).”.*

3) The X-ray structure is resolved only for the Zn-complex, did the author try the Fe-complex instead?

We would love to solve the Fe-complex crystal structure. We tried to obtain crystals of the iron complex, but we were unlucky until now. We are currently performing both crystallization under a glow box at the synchrotron facility, and soaking experiments from cobalt crystals. We hope to be successful soon.

4) In the SI, page 11, Supplementary Fig. 2 in the text should be Fig. 3; SI, page 13, Supplementary Fig. 3 in the text should be Fig. 4.

We thank the Reviewer for noting the mistakes, which have been corrected. Many apologizes for this.

5) The yield of the peptide synthesis is 65%, is this HPLC yield or isolated yield? ESI-MS spectrum should be given for the purified peptide, rather than the crude, as multiple masses can be observed in the Supplementary Fig. 11.

We apologize for the inaccuracy. We revised the text on page 19 (and in the SI) by detailing that 65 % was the yield of the isolated crude: “The isolated crude product was obtained in 65% yield (based on the resin substitution), with 50% HPLC purity.” Moreover, we acquired the LCMS spectrum of the purified peptide (p. 19-21 of the Supplementary Information).

Reviewer #2 (Remarks to the Author):

Overall, I found the described work to be exciting and relevant to the times. The authors created a novel nano-sized biological photoelectron acceptor that can be used as a tool for broader applications that require electron transport. The designed system contains a single polypeptide chain consisting of only 28 residues that contains internal 2-fold symmetry. They effectively created a protein knot to tetrahedrally coordinate metal that is more structurally stable than a Zn-finger. They performed the appropriate biophysical experiments (EPR, mass spectrometry, UV-VIs titrations and circular dichroism) to prove the designed protein coordinated iron such that it could be reduced and oxidized on demand. Then crystallized the protein with Zn²⁺ Ultimately, I would prefer the crystal structure contain iron instead of zinc, but I recommend the manuscript be published without any major changes.

We really thank the Reviewer for the inspiring words of estimation for our work. We hope that the revised version could be of even more interest to her/him.

Below is a list of minor changes that should be made.

To improve the paper, the authors should have a figure of the Cp Rd with its first and second coordination spheres defined. They draw many comparisons to this structure throughout the manuscript and it would be easier for the reader to have a picture instead of having to pull up the structure from the PDB.

We appreciate this suggestion. We added an additional figure (Figure 1) highlighting the first and second coordination sphere of V44A Cp Rd.

I found the crystal packing description on page 8 (last paragraph of section “ZnMETPsc1 crystal structure reveals a handful of secondary motifs”) to be a distraction and should be deleted. The description doesn’t lead to any relevant conclusions. I believe it was to setup the placement of the symmetry related Arg in the second coordination sphere- but is not needed.

The description has been removed as suggested.

Also Supplemental, Figure 8: This figure doesn’t add any special information to the paper and can be removed. The figure is too busy to fully understand without being able to rotate it. The placement of the Asp and Arg residues is not obvious. I assume the red balls signify water molecules.

Supplementary Figure 8 has been removed as suggested.

Figure 2, part d: add the distances for the coordinating bonds.

Figure 2d (Figure 3d in the revised manuscript) has been modified accordingly.

The rest of the requested changes are typos and bad sentence structure that can easily be corrected.

Page 2: lines 6-9; re write: “We assembled in a as small as 28-residue peptide the quintessential elements required to correctly fold around a single iron redox center, coordinated to four cysteinyl thiolates (FeCys4 site), and to efficiently function in electron- transfer.”

Page 3: line 14: put a comma after “to achieve for the first time in protein design”

Page 3: line 16: Nature should not be capitalized.

Page 3: line 23: clarify the statement about rubrerythrin.

Page 4: line 16: restate “ half-sized respect to natural Ads”

Page 4: line 25: “second sphere to second coordination sphere

Page 5: line 1 : Sulpur should not be capitalized.

Page 5, line 25: change “identifying” to “identify”

Page 6, line 4: packer??

Figure 1: legend, line 10: change “found with” to “found from”

Page 7, line 8 & page 9, line 6: change “buldge “ to “bulge”

Page 7, lines 7, 8 & 9: change “2-residues” to “2-residue”

Page 7, line 18: Sentence starting with The overall backbone should be rewritten.

Page 8, lines 11 & 12: Why is this sentence important? “In addition, interactions between Tyr16 and the equivalent residue of a crystallographic related METPsc1 molecule are observed with a 3.32 Å distance between -OH atom groups.”

Page 8: the crystal packing information is not relevant.

Page 8, lines 20 & 21: change “and containing Zn²⁺” to “that contain Zn²⁺”

Page 10, line 18: change “(one and two order of magnitude, respect to Zn²⁺ and Co²⁺, respectively)” to “(one and two orders of magnitude for Zn²⁺ and Co²⁺, respectively)”

Page 10, line 19: change “most probably attributable to “ to “most likely attributable to”

Page 12, Figure 4: define the Fe²⁺ and Fe³⁺ species in part b.

Page 19: line 5: “statistic” to “statistics”

Page 19 lines 5 & 6: “Crystals presented an orthorhombic unit cell with space group C2221. “change to something like “ Crystals grew in the orthorhombic space group C2221.”

Page 19: line 25 & Page 20, line 10: change “25°C” to “25 °C”

Page 20: line 18: Change “30% of glycerol as glassing agent to an” to “30% glycerol to an”

Page 21: line 2: move “ under argon” to “All cyclic voltammetry experiments were performed under argon with a Potentiostat...”

Page 21, line 6: change “For all the measurement” to F”or all measurements”

Page 21: lines 10 & 11: clarify “Cyclic voltammetry experiments on freely diffusing FeMETPsc1 were performed by adapting a previously published procedure, at 15 °C”

Page 22, lines 1 & 6: availability misspelled.

Authors need to define acronyms at the first instance of use: some examples:

Page 3: line 23: define SHE before using acronym

Page 4; line 7: define METP

Page 5; line 11: define Cp

Wavelength of data collection differs between main manuscript (1.00 Å) and supplementary table 2 (1.24 Å).

Resolution of refined structure also differs between main manuscript (1.44 Å) and supplementary table 2 (1.34 Å).

Items missing from supplementary table 2: redundancy of data, add the RSCC and RSR of Zn ions, add the mol probity score and clashscore, footnotes explaining the terms.

Supplemental, page 6: Define where to find the MASTER software as you did PyMOL.

Supplemental, page 7: change “grigoryanlab“ to “Grigoryan lab”

Supplemental, Page 11: (Supplementary Fig. 2) should be (Supplementary Fig. 3)

Supplemental, page 13 (Supplementary Fig. 3) should be (Supplementary Fig. 4)

Supplemental, pages 15 & 16: Change” The C-terminal strand (Strand F) folds in a head-to-tail manner to antiparallely align with the first strand, “ to “ The C-terminal strand (Strand F) folds antiparallel with respect to the first strand,”

We are profoundly grateful to the Reviewer #2 for the deep and careful reading of the manuscript. All these typos and minor issues have been fixed. Many apologizes for this.

Reviewer #3 (Remarks to the Author):

Designed Rubredoxin miniature in a fully artificial electron chain triggered by visible light by the group of Lombardi and Pavone.

The group has important and valid contributions in designing metal sites into de novo proteins, and in engineering of metalloprotein functions in designed and native scaffolds, in the field of artificial enzymes (heme proteins).

In this manuscript, the authors assembled a small peptide required to correctly fold around a single iron redox center, coordinated to four cysteinyl thiolates (FeCys₄ site), a mimetic compound for Rubredoxins that can participate and function in electron-transfer.

(Fe³⁺, Fe²⁺) METPsc1 and Zn²⁺METPsc1 were synthesized with success and the crystal structure Zn²⁺METPsc1 was detailed analysed. Fe³⁺METPsc1 was explored by spectroscopic methods (VIS, CD, EPR) and electrochemical properties determined, always with counterpoint with the ferrous form. The high

reduction potential compared to natural and designed FeCys4-containing proteins was exploited as a terminal electron acceptor of a fully artificial chain triggered by visible light.

The work is exciting and deserves to be published.

We thank the Reviewer for the nice words of appreciation to our work.

A few points to clarify:

i) The METPsc1 ligand was obtained in large amounts. Fe and Zn could be reconstituted in the template. This is a fact well known for rubredoxins that can also accept a large number of metals (mainly transition metal) and even Ga can be used as an interesting isomorphous replacement, for Fe³⁺. Why the crystal structure was extensively done in for and Zn²⁺METPsc1? I understand the spectroscopy was conducted for (Fe³⁺, Fe²⁺) METPsc, since Zn²⁺ is silent in EPR and no absorption bands are observed in Vis region (as well no redox).

We tried to obtain crystals with several metals as well as Ga³⁺ as a Fe³⁺ proxy. We were able to get diffracting crystals also with Cd²⁺ and Co²⁺, in the same crystallizing conditions of Zn²⁺. Now, we are trying to get other metals by exchanging the more labile cobalt complex. Some crystallization trials have been set up for iron under a glow box directly at the synchrotron facility, but we are still in the process of getting high resolution data. Nevertheless, we acquired very good NMR data in solution for the Zn²⁺, Co²⁺, and Ga³⁺ complexes, and we will report a deeper characterization in solution in a following manuscript.

ii) Table I compares the spectroscopic parameters of FeMETPsc1 and Cp Rd in Fe(II) and Fe(III) oxidation states. There is a clear match.

The EPR data for Fe²⁺ states should not be indicated as (-) is a S=2 state, as it looks EPR silent (difficult to be observed by EPR using different modes, and really observed by MB techniques). High-spin integer spin Fe²⁺ (S = 2) is more difficult to observe by EPR methods and low-spin Fe²⁺ is EPR silent. Both oxidation states in Rd type proteins are high-spin systems. Mössbauer (MB) spectroscopy is a particularly suitable technique for investigating the valence and spin-states of iron sites in Rd.

The Reviewer is correct in saying that Fe²⁺ is potentially amenable to EPR investigations, however, as noted by the Reviewer its integer spin state makes EPR detection at conventional frequencies (X- and Q-band) problematic due to large Zero Field Splitting (zfs). The problem can be circumvented by using High field EPR ($\nu \geq 95$ GHz), which has been successfully employed to obtain the full set of spin-Hamiltonian parameters for model systems and Cp Rd. It is worth noting that ferrous iron, despite its integer spin and typically large zfs, is not exactly “EPR silent” at conventional frequencies and in some cases X- and Q-band EPR observations have been possible at very low T and using parallel mode detection (Yoo, S., Meyer, J., Achim, C. et al. JBIC 5, 475–487 (2000). <https://doi.org/10.1007/s007750050008>). However, detection of such signals is not always possible due to the low intensity and large line widths involved. This was our case, and therefore we inserted the (-) symbol in Table 1. Following the comment of competent Reviewer 3, we modified the Table inserting appropriate references to the literature as noted above.

Table 1. Spectroscopic parameters of FeMETPsc1 and Cp Rd in Fe(II) and Fe(III) oxidation states.

		Fe ²⁺ METPsc1	Fe ²⁺ Cp Rd	Fe ³⁺ METPsc1	Fe ³⁺ Cp Rd
UV-Vis	λ/nm ($\epsilon/\text{mM}^{-1}\text{cm}^{-1}$)	311 (7.73), 331 (4.43)	311 (10.8), 333 (6.3) ⁴²	345 (7.28), 370 (8.33), 494 (6.54), 570 (3.13), 745 (0.33)	350 (7.00), 380 (7.70), 490 (6.60), 570 (3.20), 750 (0.35) ⁴³
CD	λ/nm (+/-)	312(-), 333(+)	314(-), 335(+) ⁴⁴	437(+), 502(-), 557(+), 632(-)	437(+), 500(-), 560(+), 635(-) ⁴⁴
EPR	g_{eff}	^a	^a	9.15, 4.26	9.4, 4.3 ⁴²

^a Although the high-spin (S = 2) ferrous iron is a paramagnetic species its integer spin state makes it usually “EPR silent” under normal experimental conditions. Spin-Hamiltonian parameters have been measured by means of high-frequency EPR (HFEP),

$\nu \geq 95$ GHz) in references⁴⁵⁻⁴⁶. An effective $g = 2.08 \pm 0.01$ has been reported from X- and Q-band EPR studies for a variant of Cp Rd⁴⁷. Due to low intensity and the large line widths involved, this signal was not observed in our experiments.

iii) Figure 4. FeMETPsc1 redox characterization. a, UV-Vis monitoring of Fe²⁺+METPsc1 (blue trace) aerobic oxidation to Fe³⁺+METPsc1 (red trace). blue trace or purple trace ???

We were referring to the purple trace. We apologize to the Reviewer for the mistake and thank him for pointing out our oversight. We corrected accordingly in the text.

iv) The FeMETPsc1 possesses significantly high potential value (121 mV vs SHE), exceeding the typical range for prokaryotic Rds (-100/+50 mV). The phrase could be modified ... an high reduction potential value (121 mV vs SHE), slightly higher than the values observed for Rds.

We corrected according to the Reviewer's suggestion.

v) Photoinduced electron transfer from ZnMC6* to Fe³⁺+METPsc1 is an interesting observation, as a possible reaction scheme of the synthetic electron cascade. Other couples could be explored. More details should be given on ZnMC6*.

*According to the Reviewer remark, we added more details about ZnMC6*a in the introduction (p. 4-5, l 26, 1-2): "In particular, ZnMC6*a is the Zn²⁺ derivative of MC6*a, the best performing artificial peroxidase model, able to host several metal ions (Fe, Mn, Co), displaying different activities." We also added more detail about ZnMC6*a degradation during the photoinduced electron transfer cycles (p. 17, l. 8-14 and Supplementary Fig. 9). We hope that in a near future, we and other groups will exploit both ZnMC6*a and METPsc1 as partners in redox processes. In particular, we believe that, given the simplicity of the METPsc scaffold, its engineering in biological electron transport chains could be tested for other biologically-relevant redox partners.*

Reviewer #4 (Remarks to the Author):

This manuscript describes the design and synthesis of a functional artificial rubredoxin (Rd)-like FeCys4-cluster peptide. This miniaturised metallo-‘protein’ (METPsc1) was designed de novo by dissecting the Fe-binding region from a high-resolution structure of Rd and then searching for the optimal peptide fragment that could bridge the two truncated sequences. The final model was obtained by Monte Carlo sampling. The Rd miniature was synthesised and characterised by X-ray diffraction as the Zn complex. Coordination to the metal closely mirrors that of native Rd and matches exceptionally well onto the computationally modelled structure. Fe binding was then assessed using UV-Vis, CD and EPR; a 1:1 complex is formed with the metal and the cluster gives characteristic LMCT UV-Vis bands and EPR resonances. Reversible redox cycling is also demonstrated. CV of the cluster reveals a high reduction potential relative to native Rds. Finally, Fe-METPsc1 is used as the terminal electron acceptor in a visible-light-mediated artificial electron cascade.

Based on the fact that the Fe-METPsc1 was designed from ‘scratch’ and very well characterised, including X-ray data, the novelty/originality of this paper is high. The integration of the FeMETPsc1 into an electron cascade further enhances the impact of the publication and points towards potential applications for de novo designed proteins in complex artificial systems. The methods described in the main paper, and details in the SI, are sufficiently detailed to allow reproduction of the results.

We thank the Reviewer for the encouraging words reserved to our work. We really appreciated them.

The computational design aspect of this work is outside my area of expertise so I will leave it to other reviewers to comment on the method; however, due to the extensive characterisation of the resulting METPsc1, I have no doubt regarding the effectiveness of the approach. My only real concern is in regard to Figure 6c/6d; the UV-Vis LMCT band at 496 is visible only as a flat absorbance next to the large band at around 400 nm from ZnMC6*a. Furthermore, the shift from 311 nm to 314 nm is very slight. These results are complicated by the fact that the ZnMC6*a compound absorbs at 496 nm after successive rounds of oxidation and reduction. Since the final results of this paper are based on the disappearance of the 496 nm absorbance and 311 to 314 nm shift, the authors should include the UV-Vis traces (overlaid if clear or with Abs values if not) for all time points of Figure 6d in the SI to show the gradual changes in each peak. It would be ideal if this experiment

could be repeated at a higher concentration (if that is possible) to give a more pronounced band at 496, however, this is not necessary if the UV-Vis spectra of the time points from Figure 6d are clear. Finally, it's not clear to me how the concentration values for the oxidised and reduced clusters are calculated for Figure 6d. Please add this info into the SI.

*According to the Reviewer request (see also reply to remark 2 of Reviewer 1), we performed again the experiment in Figure 6 under different conditions, and we reported it as a new Figure 6. In this experiment, we changed the ZnMC6*a:FeMETPsc1 ratio, by fixing their concentrations to 5 μ M and 40 μ M, respectively. We also kept METPsc1 peptide concentration in slight excess to be more confident that any free iron could alter or have any role in the electron transport process. Moreover, irradiation time has been kept constant to 20 minutes for each cycle, to maintain a constant light dose. Under these conditions, we are not able to reach full reduction of FeMETPsc1 in the first cycle (most probably because we decreased the ZnMC6*a:FeMETPsc1 ratio), nevertheless we were able to perform partial reduction of FeMETPsc1 for at least 4 cycles before clear bleaching of ZnMC6*a hampered further cycles to occur.*

A few additional points:

The METP abbreviation should be clearly defined on first use.

We have defined acronyms as they first appear in the text. Thank you.

For Figure 3, the four plots a, b, c, d are positioned clockwise which isn't standard formatting (compared to c below a and d below b) – a very minor point.

We were aware of this non-standard formatting, we preferred to keep it that way because this allowed us to match UV and CD data for an easier up/down comparison of the spectra. If possible, we would prefer to keep this order.

The amount of dithionite added for each redox cycle (Figure 4b) should be stated in the SI – an 'excess' is mentioned in the text, but clarity would aid reproduction of the experiments.

According to the Reviewer suggestion, we revised the related paragraph in the method section of the main text: "Next, the solution was sequentially purged with air to form the Fe³⁺ complex, then with argon and finally reduced with 0.2 μ L of 0.5 M sodium dithionite, prepared under Ar atmosphere, to restore the Fe²⁺ complex." (p. 21 l. 22-24).

Line 6-7 of the abstract ("... in a as small as...") should be re-worded for clarity.

We reworded as follows: "We assembled into a miniature 28-residue protein the quintessential elements ..."

In my opinion, this manuscript is of significant novelty and impact for publication in Nature Communications and should be accepted once the above concerns have been addressed.

We thank the Reviewer for the kind remarks, and we hope we have satisfied her/his concerns.

REVIEWERS' COMMENTS

Reviewer #1 (Remarks to the Author):

Most of my previous concerns have been nicely addressed (although I would love to see further application of this strategy for a more applicable artificial protein). Nevertheless, this reviewer agrees that the manuscript is now in a good shape for a publication in Nat. Commun.

Reviewer #2 (Remarks to the Author):

They authors have addressed all my previous concerns. I support the publication of this manuscript.

Reviewer #3 (Remarks to the Author):

"Designed Rubredoxin miniature in a fully artificial electron chain triggered by visible light".

I would like to comment, in terms of the comments made (reviewer#3), that I am satisfied with the answers. ´

In addition, I would feel more comfortable if the authors avoid in Table I (note) the term "EPR silent" and replace it with "difficult to detect by EPR" since it is an integer spin $S=2$.

Reviewer #4 (Remarks to the Author):

The authors have addressed my main points of concern in their revised manuscript. In particular, the updated figure 7C now shows the clear disappearance of the 494 nm peak and appearance of a signal at 311 nm. This data is now much clearer than in the original manuscript, leaving no room for doubt concerning this key result.

In my opinion this manuscript should now be accepted without further revision.

REVIEWER COMMENTS

AUTHOR RESPONSE

Reviewer #1 (Remarks to the Author):

Most of my previous concerns have been nicely addressed (although I would love to see further application of this strategy for a more applicable artificial protein). Nevertheless, this reviewer agrees that the manuscript is now in a good shape for a publication in Nat. Commun.

We thank the Reviewer for her/his kind words of appreciation, and we are working hard to deliver to the community an exciting set of designed metalloproteins based on this scaffold.

Reviewer #2 (Remarks to the Author):

They authors have addressed all my previous concerns. I support the publication of this manuscript.

We really thank the Reviewer for the very supporting words she/he reserved to our work. And we hope she/he may continue following our work in metalloprotein design in the future.

Reviewer #3 (Remarks to the Author):

I would like to comment, in terms of the comments made (reviewer#3), that I am satisfied with the answers. In addition, I would feel more comfortable if the authors avoid in Table I (note) the term "EPR silent" and replace it with "difficult to detect by EPR" since it is an integer spin $S=2$.

We are glad that the Reviewer is satisfied with the answers provided in the previous round of revisions, and we apologize for the misleading comment we included in the note to Table 1. We revised "EPR silent" as suggested with "usually difficult to detect under standard experimental conditions".

Reviewer #4 (Remarks to the Author):

The authors have addressed my main points of concern in their revised manuscript. In particular, the updated figure 7C now shows the clear disappearance of the 494 nm peak and appearance of a signal at 311 nm. This data is now much clearer than in the original manuscript, leaving no room for doubt concerning this key result. In my opinion this manuscript should now be accepted without further revision.

We cannot be more enthusiastic to read that now the Reviewer is fully convinced by our data. We really think that the contribution by all the reviewers significantly helped us in pushing further the quality of the manuscript and of the overall scientific soundness of the publication